# Atypical structural tendencies among low-complexity domains in the Protein Data Bank proteome

**Sean M. Cascarina**[ID]*, **Mikaela R. Elder**[ID], **Eric D. Ross**[ID]*

Department of Biochemistry and Molecular Biology, Colorado State University, Fort Collins, CO, United States of America

* Sean.Cascarina@colostate.edu (SMC); Eric.Ross@colostate.edu (EDR)

**Data Availability Statement:** All relevant data are within the manuscript and its Supporting Information files.

**Funding:** This work was supported by the National Science Foundation (https://www.nsf.gov/), grant

## Abstract

A variety of studies have suggested that low-complexity domains (LCDs) tend to be intrinsically disordered and are relatively rare within structured proteins in the Protein Data Bank (PDB). Although LCDs are often treated as a single class, we previously found that LCDs enriched in different amino acids can exhibit substantial differences in protein metabolism and function. Therefore, we wondered whether the structural conformations of LCDs are likewise dependent on which specific amino acids are enriched within each LCD. Here, we directly examined relationships between enrichment of individual amino acids and secondary structure tendencies across the entire PDB proteome. Secondary structure tendencies varied as a function of the identity of the amino acid enriched and its degree of enrichment. Furthermore, divergence in secondary structure profiles often occurred for LCDs enriched in physicochemically similar amino acids (e.g. valine vs. leucine), indicating that LCDs composed of related amino acids can have distinct secondary structure tendencies. Comparison of LCD secondary structure tendencies with numerous pre-existing secondary structure propensity scales resulted in relatively poor correlations for certain types of LCDs, indicating that these scales may not capture secondary structure tendencies as sequence complexity decreases. Collectively, these observations provide a highly resolved view of structural tendencies among LCDs parsed by the nature and magnitude of single amino acid enrichment.

## Author summary

The structures that proteins adopt are directly related to their amino acid sequences. Low-complexity domains (LCDs) in protein sequences are unusual regions made up of only a few different types of amino acids. Although this is the key feature that classifies sequences as LCDs, the physical properties of LCDs will differ based on the types of amino acids that are found in each domain. For example, the sequences "AAAAAAAAAA", "EEEEEE EEEE", and "EEKRKEEEKE" will have very different properties, even though they would all be classified as LCDs by traditional methods. In a previous study, we developed a new method to further divide LCDs into categories that more closely reflect the differences in their physical properties. In this study, we apply that approach to examine the structures

number MCB-1817622 awarded to EDR, and by the National Institute of General Medical Sciences (https://www.nigms.nih.gov/), grant number R35GM130352 awarded to EDR. The funders had no role in study design, data collection and analysis, decision to publish, or preparation of the manuscript.

**Competing interests:** The authors have declared that no competing interests exist.

of LCDs when sorted into different categories based on their amino acids. This allowed us to define relationships between the types of amino acids in the LCDs and their corresponding structures. Since protein structure is closely related to protein function, this has important implications for understanding the basic functions and properties of LCDs in a variety of proteins.

## Introduction

Protein sequence, in combination with surrounding physicochemical environment, fundamentally determines protein form and function. Protein sequences vary with respect to sequence "complexity", which is effectively a measure of the amino acid frequencies within a region of defined length. Most naturally-occurring proteins contain a rich mixture of residues drawn from the 20 canonical amino acids. However, a surprising number of proteins possess relatively simple regions, termed low-complexity domains (LCDs), which are characteristically comprised of only a subset of amino acids.

LCDs are found across a broad array of organisms [1], but they are particularly abundant in eukaryotes and participate in a variety of normal and pathological processes [2–19]. Consequently, a number of studies have examined the functions of LCDs and LCD-containing proteins. Proteins with LCDs exhibit differences in metabolic regulation, molecular interaction partners, subcellular localization, and molecular functions [18,20,21], all of which hinge upon the primary amino acid enriched in each LCD [18]. Importantly, even LCDs enriched in highly similar amino acids can be associated with specialized molecular functions and distinct trends in metabolic regulation. For example, while both N-rich and Q-rich LCDs are associated with transcription, N-rich domains are also associated with P-bodies, whereas Q-rich domains are associated with functions related to endocytosis [18]. Similarly, while enrichment of most hydrophobic amino acids is associated with low protein abundance, low protein half-life, and low translation efficiency, progressive enrichment of alanine or valine is associated with complete opposite trends in protein metabolism [18].

While the functions of LCDs have been explored in a variety of organisms, the structural characteristics of LCDs are relatively poorly understood. Early studies of LCDs indicated that they are underrepresented among structured proteins in the Protein Data Bank (PDB) [22,23], leading to the pervasive perception that LCDs are generally intrinsically disordered. Indeed the most abundant types of LCDs in eukaryotic proteomes are those enriched in polar and charged amino acids [14,15,18]. Naturally, these types of LCDs tend to adopt an ensemble of disordered conformations, which may be expanded or collapsed depending on the composition and distribution of constituent amino acids [24]. However, under certain conditions, even LCDs that favor disorder may adopt ordered conformations. For example, changes in intracellular environment (e.g. pH) might induce folding of a disordered domain, or disordered regions may adopt stable conformations when docked to folded intramolecular domains or bound to intermolecular binding partners. Additionally, LCDs with less-extreme polar/charge content may inherently favor folded conformations. Therefore, it is reasonable to expect a variety of LCDs to adopt stable structures under particular conditions or in particular sequence contexts, which is supported by recent analyses of LCDs (treated as a single class) in structured proteins [3].

A variety of methods have been developed to identify LCDs [25–30]. Most of these methods employ pre-defined statistical thresholds to classify domains as low or high complexity and demarcate LCD boundaries. While these methods are extremely valuable and have contributed

to a number of important discoveries, they also inherently possess a number of limitations [18]. In essence, all LCD detection methods are fundamentally attempting to define regions in which a small subset of the possible amino acids are present at high density. While unified by this defining feature, LCDs are really a diverse set of domains with distinct physical properties; these properties depend upon the amino acid(s) predominantly enriched in each LCD. Furthermore, most methods for defining LCDs employ a statistically-defined threshold (which may not reflect biologically-relevant criteria [18]) and do not distinguish between LCDs of moderate compositional enrichment (e.g. 40% alanine content) and LCDs of extreme compositional enrichment (e.g. 100% alanine). Thus, parsing protein domains by both the predominant amino acid enriched in each domain and its degree of enrichment may reveal trends that are obscured when LCDs are grouped without taking these compositional features into consideration.

Here, we directly examine the relationship between local amino acid composition and structural tendencies, with a particular focus on regions at compositional extremes (i.e. LCDs). Although structured proteins are strongly biased toward sequences of high complexity, LCDs are nevertheless abundant among structured proteins. Secondary structure tendencies among LCDs diverge as sequences become less complex, and the secondary structure profiles depend on both the type of amino acid enriched and its degree of enrichment. Despite the common assertion that LCDs tend to be disordered, some types of LCDs exhibit a strong preference to adopt ordered conformations. Additionally, structural tendencies diverge even for related amino acids, suggesting that physicochemically-related LCDs are not always functionally synonymous. While existing α-helix and β-sheet propensity scales perform remarkably well for sequences of moderate to high complexity, we find that these scales systematically fail for certain types of LCDs. Finally, LCD-containing proteins are associated with distinct classes of folds, dependent upon the type of LCD (i.e. the amino acid most enriched within the LCD). These observations suggest that grouping all LCDs into a single class grossly oversimplifies the structural tendencies exhibited by each type of LCD.

## Results

### A systematic survey of the PDB proteome reveals an abundance of residue-specific LCDs

Although structured proteins present in the PDB are generally biased against LCDs [22,23], this does not preclude the possibilities that: 1) certain types of LCDs (e.g. F-rich LCDs) may tend to be highly structured, and 2) LCDs may be abundant and important within structured proteins, despite the tendency to be less common than regions of higher complexity.

To systematically examine relationships between local amino acid composition and abundance within protein sequences in the PDB, we first obtained a set of non-redundant PDB protein sequences with < 90% homology (see Methods). Additionally, membrane-embedded proteins were excluded to account for differences in folding environment, which could confound downstream analyses (see S1 File for a list of all PDB chain IDs included in the final dataset). Each PDB chain was exhaustively scanned with window sizes ranging from 10 to 100 amino acids, increasing the window size by 10 with each iteration (similar to the method implemented in [18]). As mentioned previously, the essential feature underlying all methods of LCD detection is enrichment in one (or a select few) of the 20 canonical amino acids. Therefore, for each window, the percent composition for each amino acid was calculated. For any given type of amino acid, the majority of protein chains exhibit low compositional enrichment (Fig 1), indicating that most proteins are reasonably complex, which is consistent with previous studies and *a priori* expectations. However, for most amino acids, a substantial number of

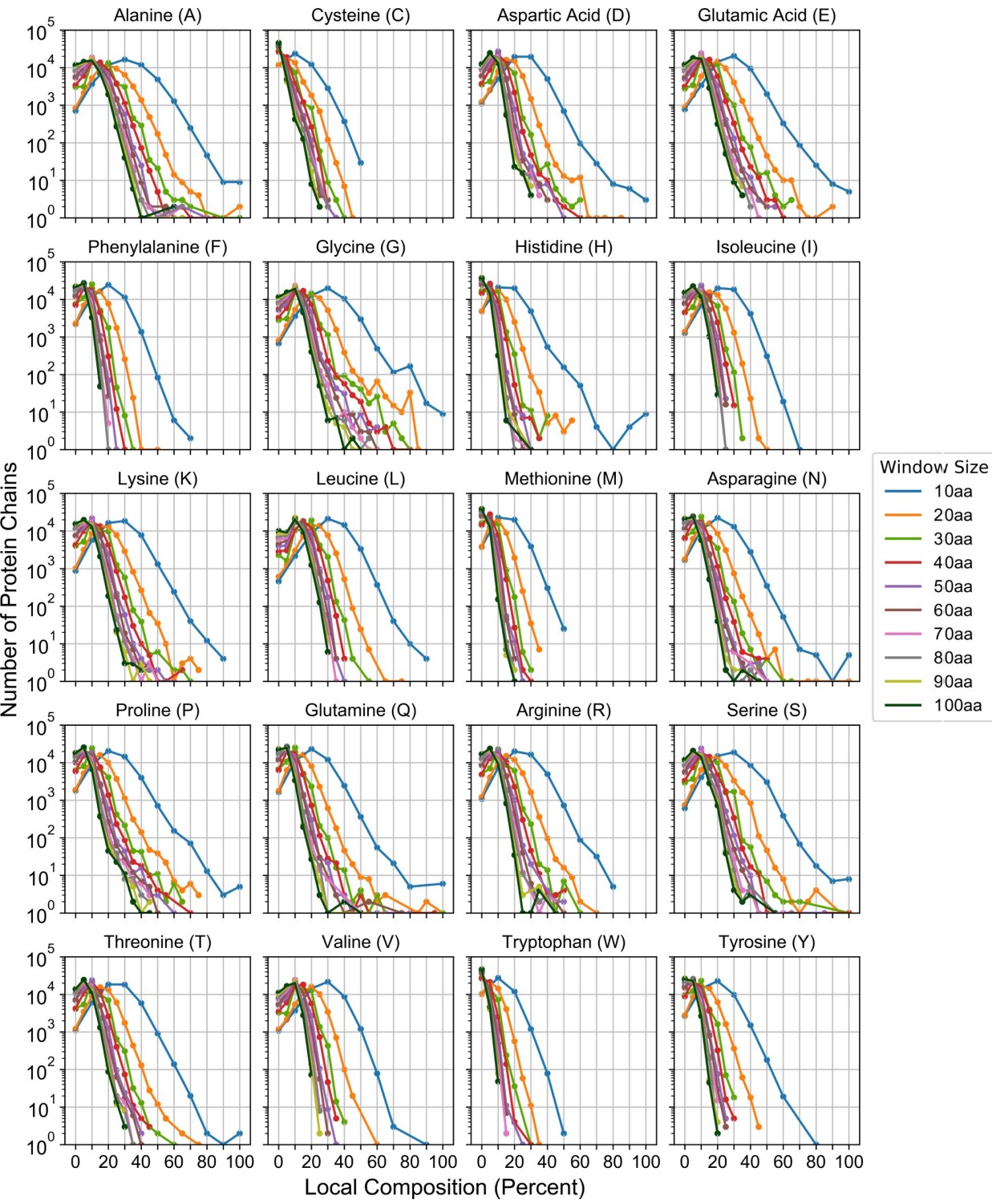

**Fig 1. LCDs are abundant in structured proteins.** Each subplot indicates the frequency distributions for all peptide subsequences within PDB proteins as a function of scanning window size and amino acid composition for each amino acid indicated in the subplot titles.

protein chains contain subsequences that achieve high compositions, particularly for smaller scanning window sizes (Fig 1). For example, for most amino acids, there are hundreds to tens

of thousands of proteins with 10-amino acid subsequences containing at least 40% composition attributable to a single type of amino acid: as a point of comparison, LCDs identified by the SEG algorithm (perhaps the most widely-used LCD identification algorithm; [25]) typically have a minimum of ~25–45% composition of the most enriched amino acid within a 12aa segment. In the most extreme case, >18k protein chains possess a region containing ≥40% leucine residues in a 10-amino acid window. Therefore, LCDs are quite abundant in PDB sequences despite the overall bias toward sequences of higher complexity.

To directly compare the frequencies in Fig 1 with frequencies obtained using traditional methods of identifying LCDs, we employed a modified version of the SEG algorithm. Briefly, PDB sequences were scanned using the default 12aa window size, and all segments scoring ≤2.2 bits were classified as LCD segments. By default, the SEG algorithm does not distinguish between LCDs of different types (i.e. LCDs strongly enriched in different amino acids). Therefore, we modified the SEG algorithm to parse LCDs into amino acid-specific categories based on the most frequent amino acid within each LCD, as indicated in [18] (also described in the Methods section). This modification allows us to dissect specific relationships between LCD types and their corresponding frequencies.

Distinct classes of LCDs exhibit markedly different frequencies within the PDB and within eukaryotic proteomes (Fig 2). While A-rich and L-rich LCDs tend to be the most common individual LCD types among PDB proteins (Fig 2A), LCDs enriched in charged or polar residues occupy most of the subsequent top-ranked positions with respect to frequency. When LCDs are combined based on general physicochemical categorization, LCDs enriched in a hydrophilic amino acid are the most abundant class both in terms of the number of PDB proteins containing a hydrophilic LCD (albeit very slightly; Fig 2A, *inset*) and the total number of regions classified as hydrophilic LCDs (Fig 2B, *inset*). To compare LCD frequencies in the PDB to LCD frequencies among whole proteomes, identical analyses were performed for the yeast and human proteomes (Fig 2C and 2D). In general, the percentage of proteins with LCDs is higher among the yeast and human proteomes compared with the PDB for nearly every type of LCD, indicating that the PDB is selectively biased against sequences with LCDs both globally and for most types of LCDs individually (which is consistent with previous analyses [3]). Proteins with hydrophilic LCDs are far more common than proteins with hydrophobic LCDs in the yeast proteome, both in terms of individual amino acids (Fig 2C) and when grouped (Fig 2C, *inset*), and this margin is much larger in the yeast proteome than in the PDB. With respect to LCD frequencies, the PDB more closely resembles the human proteome (Fig 2D), although minor differences are also apparent. For example, the margin between proteins with hydrophilic LCDs and proteins with hydrophobic LCDs is slightly larger in the human proteome compared to the PDB, although this margin is not as dramatic as that observed for the yeast proteome. Proteins with G-rich LCDs are ~2-fold higher (in terms of percentage of proteins with LCDs) in the human proteome compared to the PDB, while proteins with P-rich LCDs are ~7-fold higher in the human proteome compared to the PDB. Since both the size of the proteomes and the lengths of LCDs may differ across the different proteomes, we also evaluated the percentage of each type of residue found within the corresponding class of LCD (e.g. the percentage of alanine residues within A-rich LCDs). This effectively allows for a normalized comparison across proteomes with respect to enrichment of each type of amino acid within LCDs vs. non-LCDs. Nevertheless, the results (Fig 2E) are largely in agreement with those observed in Fig 2A–2D, indicating that both the percentage of proteins with each type of LCD and the percentage of each type of residue found within its corresponding class of LCDs robustly describe LCD enrichment across the examined proteomes.

Overall, these data highlight the abundances of LCDs when distinguished on the basis of the predominant amino acid enriched in each LCD within the PDB and certain eukaryotic

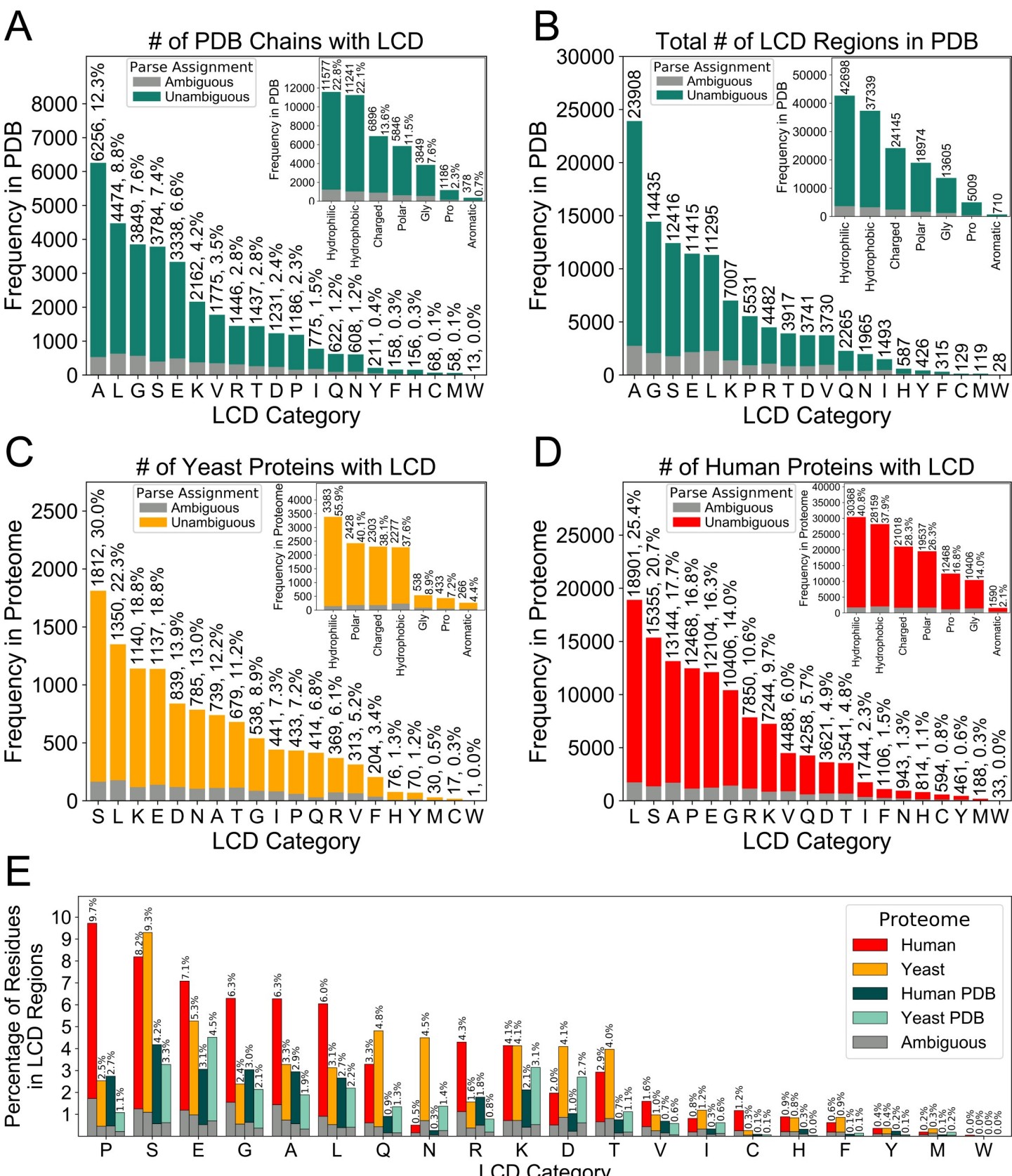

**Fig 2. Abundances of classically-defined LCDs within the PDB and eukaryotic proteomes.** The PDB proteome was exhaustively scanned using a 12aa window size, and the Shannon entropy was calculated for each segment. Segments with Shannon entropy ≤2.2 bits were classified as LCDs. For each amino acid, LCDs were assigned to that category if the frequency of the amino acid was ≥ the maximum frequency for all amino acids within the LCD sequence. The total number of PDB chain sequences with each type of LCD (A) and the total number of regions scoring as LCDs for each LCD category (B) are indicated. The number of proteins with each type of LCD were similarly plotted for the yeast and human proteomes and indicated in (C) and (D) respectively. Additionally, the percentage of each amino acid found within LCDs of that amino acid type is indicated in (E) for all 20 amino acids. For all plots, the LCDs for which the indicated amino acid was clearly the predominant amino acid in the LCD sequence represent "unambiguous" LCDs, while LCDs for which another amino acid was equally abundant within the LCD sequence represent "ambiguous" LCDs. Proteins were preferentially assigned to the "unambiguous" category if they contained at least one region that could be unambiguously identified as an LCD of a given type. Insets in the upper-right corner of panels A-D indicate the frequencies of LCDs when grouped by physicochemical categories: hydrophobic (A, I, L, M, and V); charged (D, E, H, K, and R); polar (C, N, Q, S, and T); aromatic (F, W, and Y); and hydrophilic (combination of charged and polar classes). Total values corresponding to broad physicochemical categories in the insets do not represent the sum of the frequencies of individual types of LCDs, since some proteins contain multiple distinct types of LCDs that fall into the same physicochemical category.

proteomes. If LCDs are treated as a single class, broad trends in structural tendencies will be heavily biased by how common each type of LCD is in PDB sequences, even though LCDs of different types may exhibit dramatically different structural tendencies and behavior.

## Stratification of LCDs by degree of amino acid enrichment reveals dose-dependent relationships between composition and structural tendencies

We were particularly interested in secondary structure tendencies corresponding to short regions of high density of one or a few amino acid types (i.e. LCDs). By default, the SEG algorithm uses a 12-amino acid scanning window. Therefore, for consistency with subsequent analyses, we exhaustively re-scanned all PDB sequences with a 12aa scanning window and binned each subsequence based on the composition of each amino acid within each subsequence (Fig 3). For each subsequence, the fraction of each secondary structure type was also calculated and binned with the corresponding subsequence. The fraction of each secondary structure type was then averaged across all subsequences within each composition bin (Fig 3) and plotted (Fig 4; see Methods for additional description). For most amino acids, subsequences at extreme compositions are overwhelmingly disordered (Fig 4; light grey line), consistent with previous observations. However, for a number of amino acids (particularly hydrophobic and aromatic residues), subsequences maintain a substantial mean fraction of alternative secondary structures. For example, A, F, I, L, M, V, W, and Y are predominantly associated with α-helix or β-sheet preference at their respective extremes of enrichment. While the degree of enrichment achieved by many of these residue types are often not as extreme as the degree of enrichment for polar/charged residues, these subsequences are nevertheless typically classified as LCDs. It is important to note that LCD sample sizes tend to decrease as residue count increases beyond ~3–4 residues for any given amino acid (as shown in Fig 1), which can result in spuriously large changes in secondary structure proportions at high residue count bins for each amino acid: therefore, analogous bar charts indicating sample sizes for each amino acid across all residue count bins are indicated in S1 Fig for comparison.

A detailed inspection of the composition-to-structure tendencies reveals more nuanced differences even between amino acids of very similar types. In order to facilitate a direct visual comparison between related amino acids, the mean fraction of each type of secondary structure at increasing composition levels for each amino acid (from Fig 4) were linearly interpolated and indicated as dynamic stacked bar charts in S1 Movie. For ease of comparison, amino acids were clustered based on physicochemical properties.

For regions entirely lacking the amino acid of interest ("0 Residues of Interest" frame), secondary structure tendencies do not differ dramatically, and each category roughly approximates the whole-PDB tendencies. Regions lacking P or G have the highest proportions of α-helix and β-sheet, and these proportions drop to among the lowest within the first shift

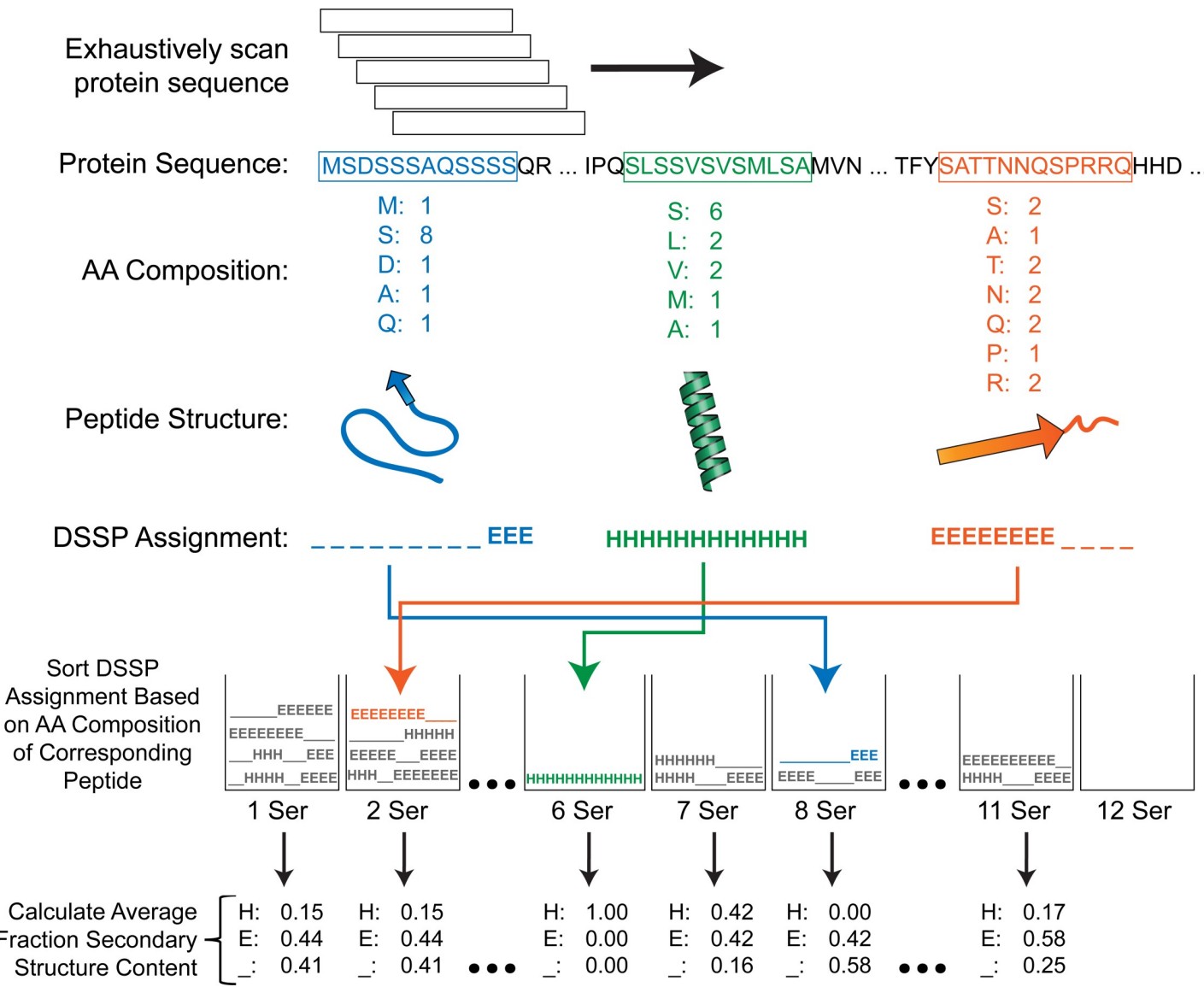

**Fig 3. Depiction of computational strategy for relating local amino acid composition to secondary structure annotations across the PDB proteome.** For each amino acid, PDB sequences were scanned with a 12aa window size. For each peptide subsequence, the corresponding DSSP (i.e. secondary structure) annotations were sorted into bins based on the frequency of the amino acid of interest (e.g. serine, in the depicted example). After an exhaustive scan of the PDB proteome, the mean fraction of each secondary structure type was calculated within each residue count bin.

(domains with only one P or G), consistent with the known role of these residues in disfavoring or precluding these secondary structure types (see, e.g., [31] and [32] for comparisons of multiple α-helix and β-sheet propensity scales, respectively). Mean α-helix and β-sheet content continues to decline precipitously as P content increases, while the disordered and irregularly-structured fractions increase. Additional dose-response relationships are apparent between composition and secondary structure for many other amino acids. With respect to physico-chemically-related groups of amino acids, differences in secondary structure tendencies begin to emerge with as few as 3 residues of interest and often become more pronounced as composition continues to increase. For example, among hydrophobic residues, V-rich regions tend to have a higher proportion of β-sheet content relative to regions enriched in other

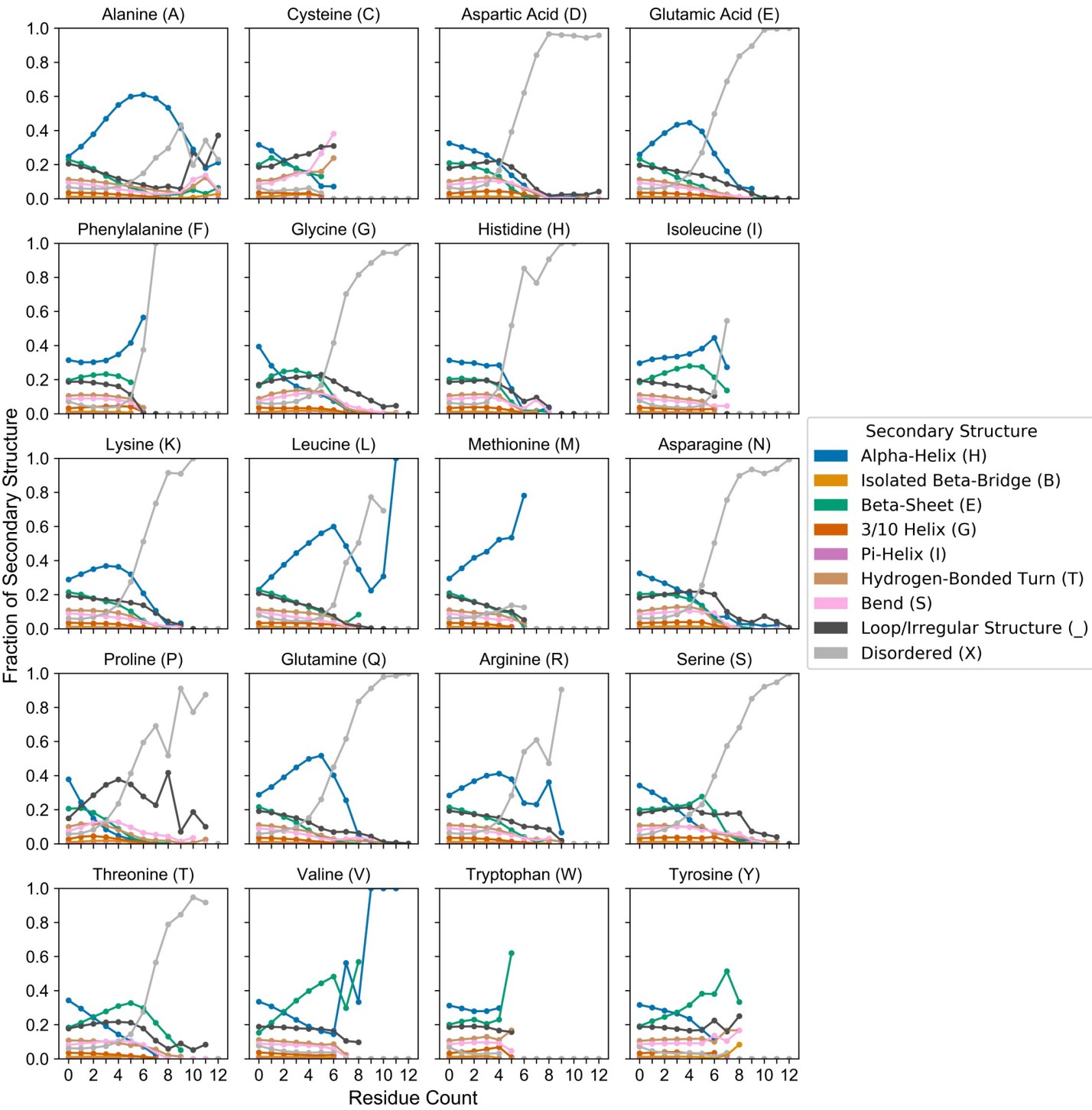

**Fig 4. Conformational tendencies are highly dependent on both amino acid type and degree of enrichment.** The PDB proteome was exhaustively scanned using a 12aa window size as depicted in Fig 3. Secondary structure tendencies across a range of enrichment for each amino acid are depicted in separate subplots. Within each subplot, scatter points represent the mean fraction of each secondary structure type across all peptide sequences within each indicated "residue count" bin. Sample sizes for all amino acids and residue count bins are indicated in S1 Fig.

hydrophobic residues, and this disparity progressively increases from 3–6 residues of interest. Regions with higher enrichment of these types of residues are rare or absent within the PDB,

resulting in spuriously large apparent changes in secondary structure tendencies. Similar trends are apparent for other groups of amino acids. Among aromatic residues, mean α-helix content is highest in F-rich regions, moderate in W-rich regions, and low in Y-rich regions, with a concomitant inverse relationship with respect to β-sheet content. Q-rich regions tend to maintain a higher fraction of α-helix content and lower fraction of β-sheet content compared to other uncharged polar residues, and this divergence progressively increases at higher compositions. Interestingly, though N and Q are often considered synonymous, previous studies have indicated that N-rich LCDs are more prone to form β-sheet-rich amyloid aggregates compared to Q-rich LCDs [33,34]. Together with our observations, this suggests that the higher β-sheet propensity for N-rich domains *vis-à-vis* Q-rich domains may reflect the biophysical behavior of these classes of LCDs in structured proteins generally (not just in the context of amyloid aggregation). Finally, among the charged residues, H-rich and D-rich regions tend to be associated with a higher mean fraction of disorder and lower fraction of α-helix content relative to E-rich, R-rich, and K-rich regions; this trend begins to develop at milder levels of enrichment but is most apparent between 5–7 residues of interest.

To verify that these results are not a consequence of using a high homology threshold (90%) to obtain the non-redundant set of PDB sequences, we also analyzed a non-redundant set of PDB sequences with <40% sequence homology. Indeed the trends in secondary structure tendencies among the 40% non-redundant PDB dataset (S2 Fig) are nearly indistinguishable from those observed in Fig 4 for the 90% non-redundant PDB dataset, indicating that our original observations are not heavily influenced by multiple instances of specific proteins with high sequence homology and a low-complexity domain.

From these observations we conclude that, while LCDs as a single class are often associated with intrinsic disorder, certain types of LCDs exhibit clear preferences for particular secondary structures, and these structural tendencies are dependent upon the degree of enrichment for each residue type.

## Direct comparison of compositional enrichment to traditionally-defined LCDs

To directly compare compositional enrichment with a set of LCDs defined using traditional methods, we again implemented our modified version of the SEG algorithm, which parses LCDs into types based on the most strongly-enriched amino acid. To gain an initial point of reference, mean secondary structure content for the entire set of non-redundant PDB chain sequences was first calculated. On average, α-helix content constitutes the highest fraction of structured proteins, followed by β-sheet content and non-disordered "loop or other irregular structure" content (Fig 5). Notably, the mean fraction of intrinsic disorder constitutes ~5–10% of all secondary structure annotations. In contrast, LCDs as a single class are associated with a greater degree of intrinsic disorder and concomitant compaction of the mean fraction of every other secondary structure type except α-helix content (Fig 5). However, more importantly, secondary structure profiles differ considerably when LCDs are parsed into sub-classes based on the predominant amino acid. These differences closely mirror those observed when using our unbiased, composition-centric approach. For example, Q-rich LCDs almost exclusively prefer α-helical or disordered conformations, whereas N-rich LCDs exhibit lower α-helix content, higher β-sheet content, and a higher fraction of residues adopting non-disordered "loop or irregular structure". A remarkably similar trend is observed among their negatively charged counterparts (E-rich and D-rich LCDs, respectively), suggesting that the slightly longer side chains of Q and E (rather than their terminal functional groups) explain these differences. Strikingly, all three main aromatic-type LCDs exhibit remarkably low intrinsic disorder

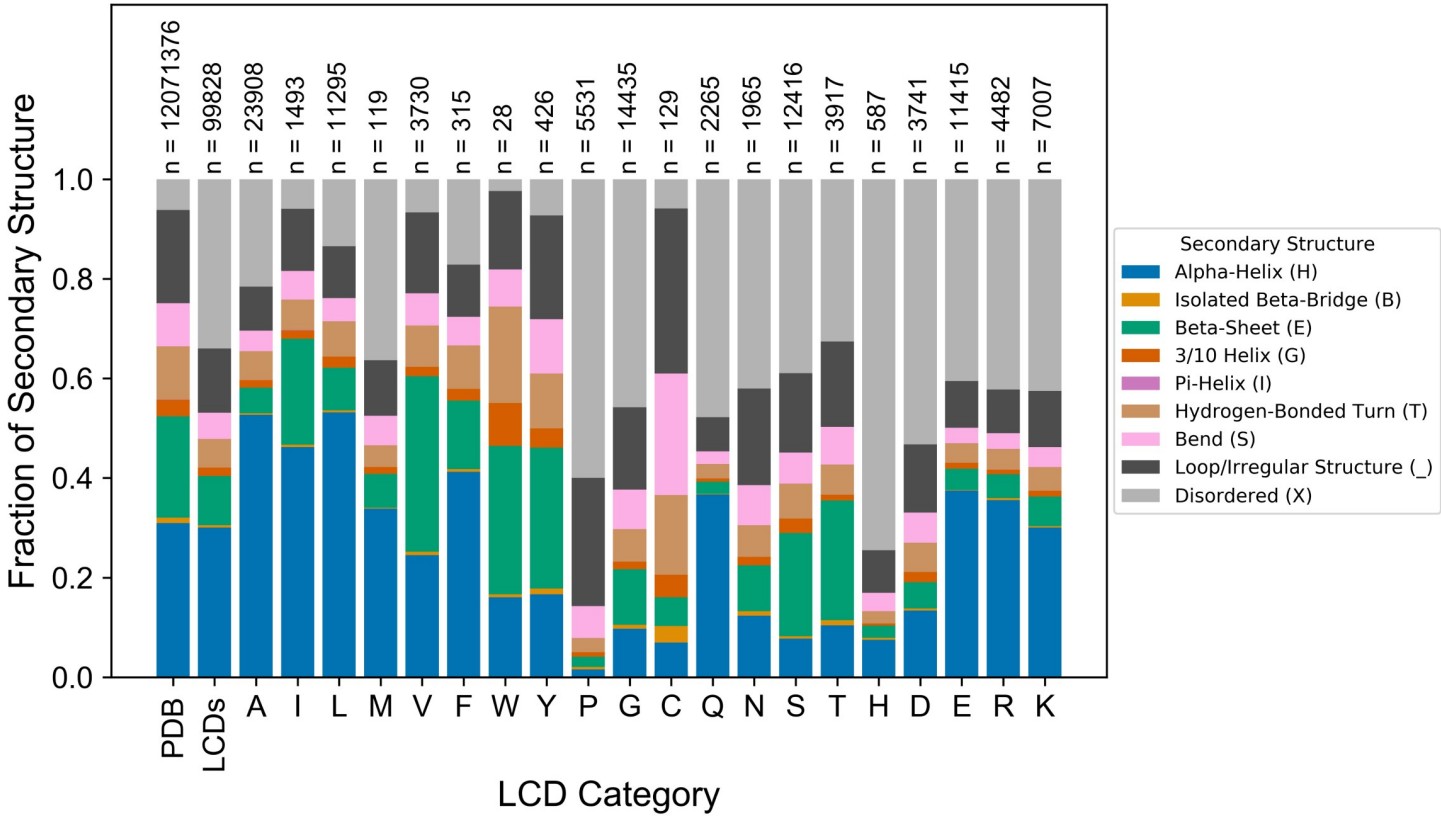

**Fig 5. LCD classes parsed by predominant amino acid exhibit unique structural tendency profiles.** LCDs with Shannon entropy ≤ 2.2 bits were parsed into amino acid categories based on the most frequent amino acid within each peptide subsequence. Bars indicate the mean fraction of each secondary structure type for all subsequences within each amino acid-specific LCD bin. For comparison, the mean fraction of each type of secondary structure for all subsequences across all proteins ("PDB" group) and for all peptides qualifying as LCDs combined into a single category ("LCDs" group) are also shown.

content, roughly equal to that of structured proteins in general ("PDB" group), further emphasizing that the relationship between low sequence complexity and intrinsic disorder is heavily dependent on LCD type. Again, these results are nearly identical to those obtained using a non-redundant set of PDB proteins with <40% sequence homology (S3 Fig).

While these data are largely in agreement with the data in Fig 4 and S1 Movie, they highlight an important limitation of grouping LCDs with no regard for composition. Parsing LCDs based on the most-enriched amino acid resolves the issue of combining physically disparate domains but does not specifically relate secondary structure tendencies to the degree of enrichment for the indicated amino acid. For example, the class of N-rich LCDs include domains that contain ~25–35% N up to 100% N. As indicated in Fig 1, sequences with lower percent composition tend to be much more abundant within the PDB, indicating that these sequences would disproportionately weight the secondary structure tendencies within each LCD category. Furthermore, using a strict, arbitrarily-defined complexity threshold does not indicate the degree of enrichment required to observe the initial development of secondary structure preferences, or how these preferences change at different levels of enrichment. Therefore, although traditional methods for defining LCDs may still be sufficient to yield important structural insights, further resolving LCDs based on degree of enrichment for the predominant amino acid allows for elucidation of dose-dependent relationships between the amino acid composition of LCDs and their corresponding structural tendencies.

## Secondary structure tendencies among highly-enriched LCDs systematically deviate from secondary structure propensity scales

A variety of statistical and experimental approaches have been used to develop α-helix and β-sheet propensity scales for the 20 canonical amino acids. However, these scales are nearly always developed using a set of structured proteins which are skewed toward high-complexity sequences, yet LCDs often exhibit unusual biophysical behavior [24]. A simplified interpretation of the secondary structure propensity scales might suggest that as LCDs approach 100% composition of a single type of amino acid (i.e. homopolymers), the α-helix and/or β-sheet propensity of that domain as a whole should approach the exact propensity value for the predominant amino acid within the LCD. However, at present, it is unclear whether these scales extrapolate well to LCDs.

To directly assess the applicability of current secondary structure propensity scales to LCDs in structured proteins, we first obtained a set of highly-enriched LCDs for each amino acid type by combining all previously-identified 12aa LCDs with ≥50% of the composition attributable to that amino acid. Therefore, in principle, the predominant amino acid in each type of LCD should exert the greatest influence on the overall secondary structure propensity of the LCD. Additionally, all windows comprised of <50% that amino acid were pooled and retained as a "non-LCD" set for comparison. Note that, in this context, non-LCDs simply refer to windows having fewer than 6 residues (out of 12) of the amino acid under consideration (e.g. windows with <6 alanines, when alanine is the amino acid under consideration). For each class of highly-enriched LCDs (defined by the amino acid that constitutes ≥50% of each LCD sequence in the class), the fraction of the defining residue annotated as α-helix or β-sheet (out of the total frequency of the defining residue among that class of LCDs) was calculated separately. For example, among A-rich LCDs, the fraction of alanine residues annotated as α-helix and the fraction of alanine residues annotated as β-sheet were calculated. This procedure was repeated for all classes of highly-enriched LCDs and the corresponding classes of non-LCD regions, then plotted against each of seven α-helix propensity scales [31,35–38] and nine β-sheet propensity scales [32,35,36,39–42] in a pairwise fashion (computational workflow depicted in Fig 6; see Methods for additional details).

In general, a moderate correlation is observed between the LCD-specific fraction of residues in α-helices or β-sheets and established α-helix or β-sheet scales (S4–S6 Figs). Correlation coefficients for the highly-enriched LCDs when compared to α-helix scales range from 0.52 to 0.78, while correlation coefficients for the highly-enriched LCDs when compared to β-sheet scales span a much broader range of 0.37 to 0.82 (S4–S6 Figs). By comparison, correlation coefficients tend to be much higher when considering the fraction of each type of residue among the corresponding non-LCD regions annotated as α-helix or β-sheet (S6–S8 Figs). When sorted by average correlation coefficient across all pairwise comparisons, the non-LCD category achieves the highest rank for both α-helices and β-sheets (S6 Fig), indicating that our derived statistic is actually in better agreement overall with all of the established propensity scales than any of the previously-derived scales. Conversely, the LCD category achieves the absolute lowest rank among the α-helix propensity scales and the third-to-last rank among β-sheet propensity scales (S6 Fig). Therefore, while established secondary structure propensity scales work quite well for non-LCD regions, they consistently perform relatively poorly for LCD regions.

For most classes of LCDs, the fraction of the defining residue in α-helices or β-sheets is somewhat consistent with propensity values derived from α-helix and β-sheet scales respectively. However, a detailed examination of the correlation plots indicates that LCDs enriched in a few specific amino acids diverge in a consistent manner from each of the secondary

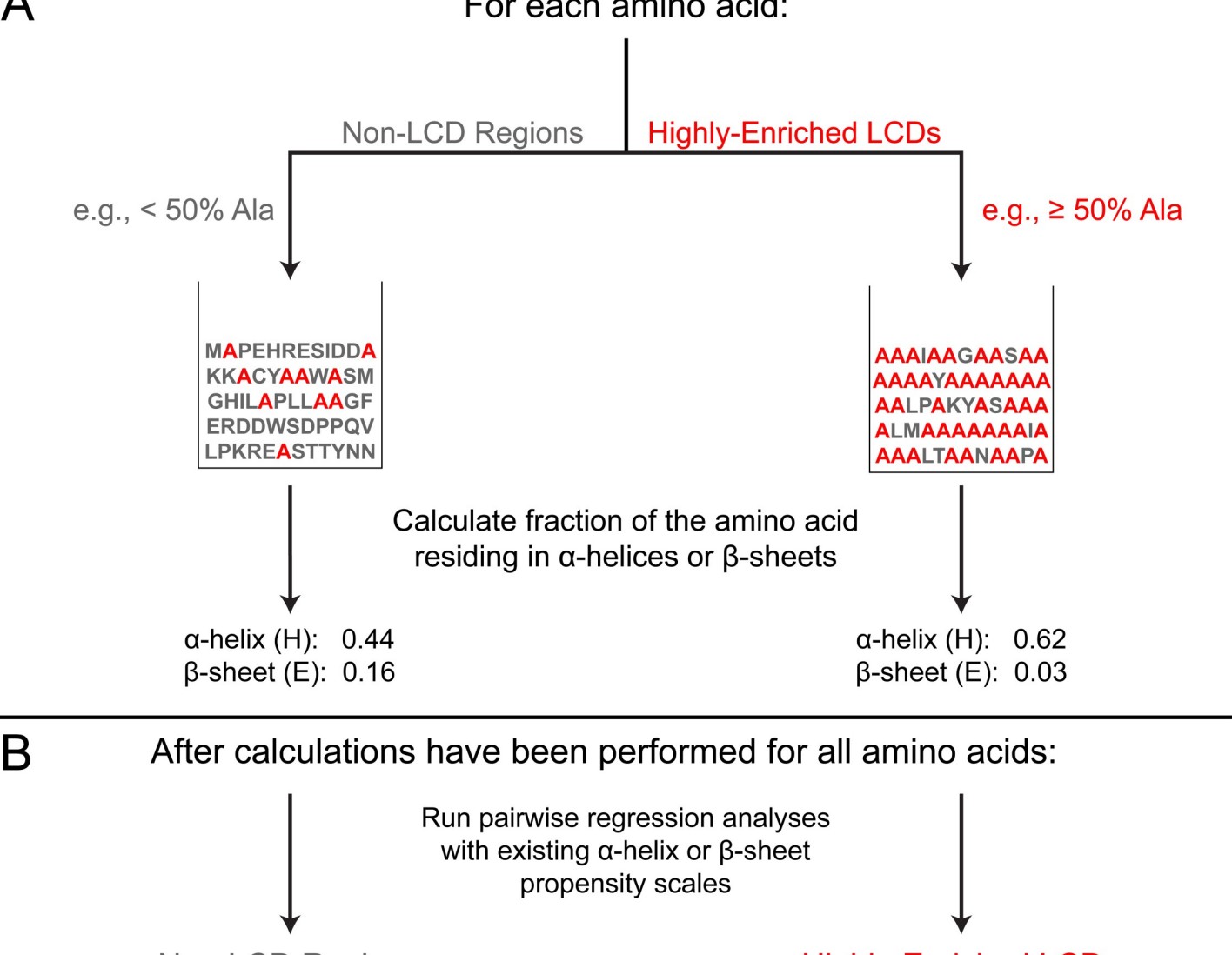

**Fig 6. Computational strategy for assessing the efficacy of established secondary structure propensity scales in predicting secondary structure tendencies among LCDs and non-LCD regions.** (A) For each amino acid, the PDB proteome is scanned using a 12aa window, and all windows are parsed into either a "highly-enriched LCD" category (windows with ≥50% composition of that amino acid) or a "non-LCD" category (in this context, defined as windows with <50% composition of that amino acid). For both categories, the fraction of that amino acid annotated as α-helix and the fraction annotated as β-sheet are calculated. This procedure is repeated for all 20 canonical amino acids. (B) Then, pairwise regression analyses are performed between the fraction of residues in α-helices and each of the α-helix propensity scales. To determine how well secondary structure propensity scales apply to LCD and non-LCD regions, regression analyses are performed separately for "highly-enriched LCDs" and the "non-LCD" category. This process is repeated for the fraction of residues in β-sheets and each of the β-sheet propensity scales. In regression analyses, missing amino acids indicate that the amino acid was removed from analyses either because few of the established secondary structure propensity scales scored that amino acid or, in the case of highly-enriched LCDs, that few LCDs with ≥50% of that amino acid exist in the PDB (see Methods).

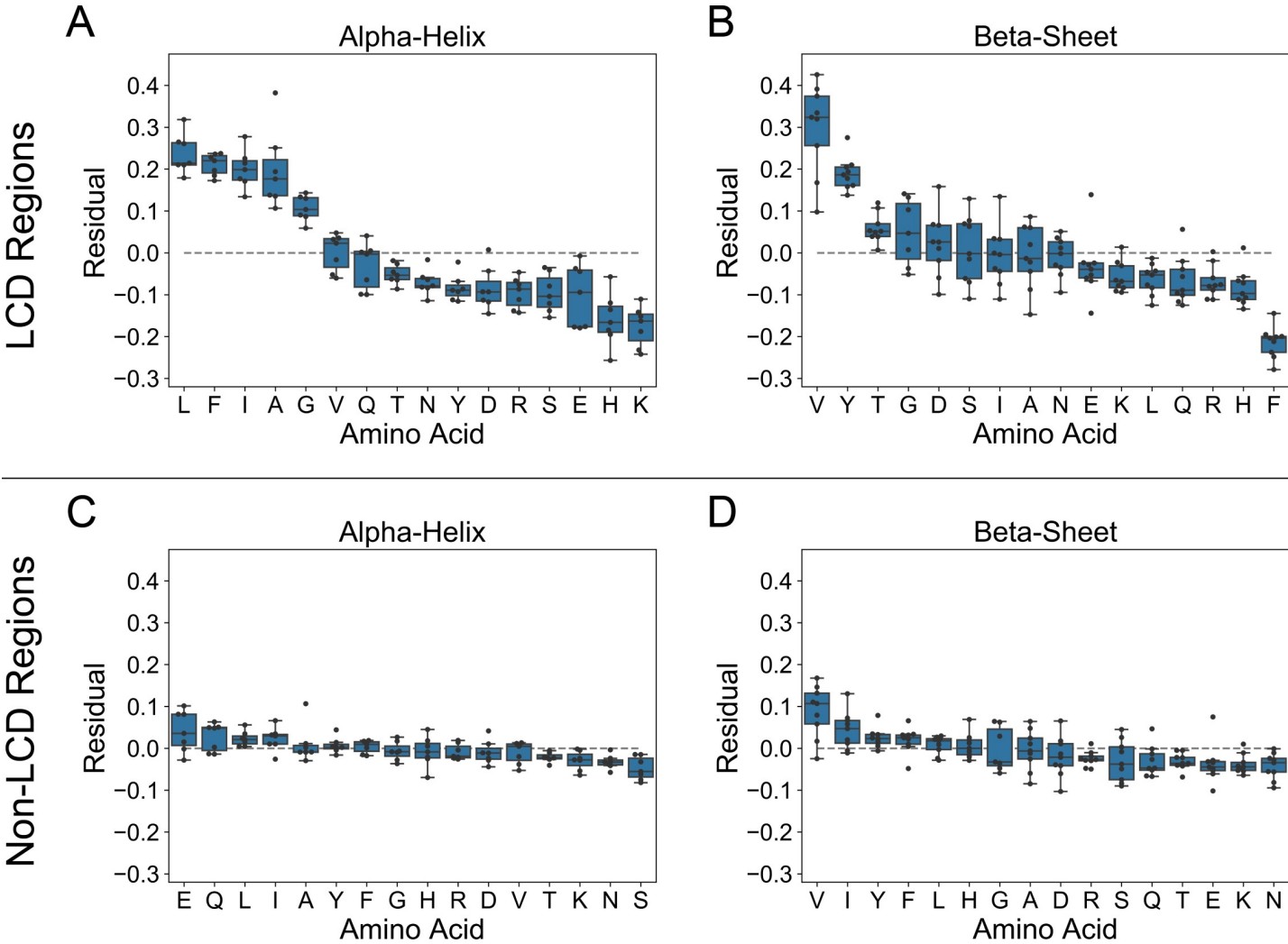

**Fig 7. Observed secondary structure content for some LCDs systematically deviates from secondary structure propensity scales.** For each set of highly-enriched LCDs (defined as those for which a single amino acid comprises at least 50% of the overall composition), the fractions of the defining amino acid in α-helices or in β-sheets were calculated separately and plotted against all secondary structure propensity scales in a pairwise fashion (see S4 and S5 Figs). For each amino acid, the residuals were calculated from the regression line and indicated in the box plots for α-helix propensity scales (A) and β-sheet propensity scales (B). For comparison, the same procedure was performed for each amino acid among non-LCD regions, and the resulting residuals are indicated in boxplots for α-helix propensity scales (C) and β-sheet propensity scales (D).

structure propensity scales (S4 and S5 Figs). In order to quantitatively assess these tendencies, the vertical displacement from the line of best fit (i.e. the residual) was calculated for each amino acid within each regression plot for the highly-enriched LCDs and the non-LCD regions. The residual essentially represents both the direction and magnitude by which structural tendencies among LCDs tend to deviate from predicted values according to established scales. With respect to α-helices, LCDs highly-enriched in L, F, I, A, or G tend to contain a high fraction of that residue in α-helices compared to their predicted values based on regression with established scales (Fig 7A). Notably, the residual values for these classes of highly-enriched LCDs are positive for all α-helix scales examined, indicating that the observed α-helix content for the defining residue is unanimously higher than expected across all α-helix scales. Conversely, LCDs enriched in K, H, E, S, R, D, Y, N, or T tend to have a low fraction of the defining residue in α-helices relative to their expected values. For the β-sheet propensity

scales, LCDs enriched in V, Y, or T tend to have a high fraction of that residue in β-sheets relative to their expected fraction of β-sheet content, while a cluster of certain LCDs (including F, H, R, Q, L, K, and E type LCDs) typically have a low fraction of the defining residue in β-sheets relative to predicted β-sheet propensity (Fig 7B). For many of the LCD classes, we observe a dose-dependent relationship between the amino acid composition and the mean residual value across all α-helix and β-sheet propensity scales (see S9 and S10 Figs and S3 File for extensive analysis and associated supplemental discussion). Additionally, complete compositional analysis of the highly-enriched LCDs indicates that the structural tendencies exhibited by each class of LCDs are not substantially influenced by a latent preference for another amino acid as the second most common residue (S11 Fig). Importantly, large residual values are not obtained when regression analyses are performed for non-LCD regions (Fig 7C and 7D).

Overall, these data suggest that while secondary structure propensity scales perform reasonably well when assessing the structural tendencies of non-LCD regions, the structural tendencies among specific classes of LCDs consistently deviate from existing α-helix and β-sheet propensity scales.

## Specialized LCDs preferentially occur within proteins that adopt particular folds

While LCDs are more often intrinsically disordered relative to sequences of higher complexity, it is clear that many of the specialized LCD types are capable of (if not prefer) adopting stable secondary structures. It is possible that structured LCDs are favored within particular 3D protein fold families, effectively occupying a protein structure niche. To examine this possibility, Pfam fold annotations mapped to PDB entries were obtained from the PDBfam database [43]. For each category of specialized LCD type (as defined by the modified SEG algorithm), the frequency of each protein fold annotation associated with the specialized LCDs was compared to the frequency of the same annotation within all PDB entries included in our non-redundant PDB proteome using Fisher's exact test with Holm-Šidák correction for multiple hypothesis testing (see Methods section for additional details). For each type of LCD, the top 10 significantly enriched Pfam annotations are indicated in Table 1 (complete statistical analyses are provided in S4 File). For some well-characterized types of LCDs, enriched Pfam annotations correspond to expected types of folds. For example, C residues are often involved in coordination of Fe-S clusters, and C-rich domains are associated with proteins containing Fe-S binding domains. L-rich LCDs are associated with a number of distinct leucine-rich repeat domain (LLRD) proteins. Collagen proteins are textbook examples of P-rich proteins, and proteins with P-rich LCDs identified in our analyses are associated with proteins containing collagen triple helix repeats.

However, aside from these well-known associations, our data implicate additional relationships between LCDs and tertiary structure families. For example, both G-rich and V-rich LCDs are significantly associated with zinc-binding dehydrogenases and alcohol dehydrogenase GroES-like domains. N-rich and I-rich domains are associated with clostridium neurotoxin proteins. W-rich domains are associated with multiple glycosyl hydrolase families, as well as reverse transcriptase and RNaseH families. S-rich and T-rich LCDs are associated with immunoglobin V-set domains and immunoglobulin C1-set domains; interestingly, immunoglobulin V-set and immunoglobulin C1-set domains tended to be the most strongly underrepresented Pfam annotations among nearly all other types of LCDs (S4 File), indicating that proteins containing these folds avoid most types of LCDs yet specifically allow for S- and T-rich LCDs. In addition to the shared associations (either between or within a LCD type), many of the enriched Pfam classes are uniquely related to a particular type of LCD. Collectively,

**Table 1. Top 10 significantly enriched Pfam annotations associated with each LCD category.** For each LCD class, up to 10 significantly enriched (lnOR > 0.0) Pfam annotations are indicated in ascending order according to Holm-Šidák corrected *p*-value.

| LCD Type | Annotation 1 | Annotation 2 | Annotation 3 | Annotation 4 | Annotation 5 | Annotation 6 | Annotation 7 | Annotation 8 | Annotation 9 | Annotation 10 |
|---|---|---|---|---|---|---|---|---|---|---|
| A-rich LCDs | Bacterial extracellular solute-binding protein (lnOR = 0.75; Adj. p = 0.0023) | | | | | | | | | |
| C-rich LCDs | Metallothionein (lnOR = 6.45; Adj. p = 3.36e-13) | 2Fe-2S iron-sulfur cluster binding domain (lnOR = 4.23; Adj. p = 1.93e-09) | Peptidase family C1 propeptide (lnOR = 5.73; Adj. p = 1.06e-07) | Spider insecticidal peptide (lnOR = 6.65; Adj. p = 0.00031) | [2Fe-2S] binding domain (lnOR = 4.32; Adj. p = 0.0005) | Papain family cysteine protease (lnOR = 3.39; Adj. p = 0.0006) | Phlebovirus glycoprotein G2 (lnOR = 5.67; Adj. p = 0.0013) | Insulin-like growth factor binding protein (lnOR = 5.35; Adj. p = 0.0022) | CO dehydrogenase flavoprotein C-terminal domain (lnOR = 3.96; Adj. p = 0.027) | Aldehyde oxidase and xanthine dehydrogenase; a/b hammerhead domain (lnOR = 3.92; Adj. p = 0.028) |
| D-rich LCDs | Type III restriction enzyme; res subunit (lnOR = 2.22; Adj. p = 0.0021) | | | | | | | | | |
| E-rich LCDs | Elongation factor Tu domain 2 (lnOR = 1.18; Adj. p = 0.0016) | NAD-dependent DNA ligase adenylation domain (lnOR = 2.19; Adj. p = 0.047) | | | | | | | | |
| F-rich LCDs | Orotidine 5'-phosphate decarboxylase / HUMPS family (lnOR = 3.03; Adj. p = 0.0013) | Influenza RNA-dependent RNA polymerase subunit PB2 (lnOR = 3.62; Adj. p = 0.019) | Eukaryotic translation initiation factor 3 subunit 8 N-terminus (lnOR = 4.7; Adj. p = 0.044) | | | | | | | |
| G-rich LCDs | Cyclophilin type peptidyl-prolyl cis-trans isomerase/CLD (lnOR = 1.47; Adj. p = 2.8e-06) | Berberine and berberine like (lnOR = 1.89; Adj. p = 2.14e-05) | Pyridoxal-phosphate dependent enzyme (lnOR = 1.1; Adj. p = 3.05e-05) | Zinc-binding dehydrogenase (lnOR = 1.02; Adj. p = 0.00022) | FtsZ family; C-terminal domain (lnOR = 1.95; Adj. p = 0.0005) | Pyridine nucleotide-disulphide oxidoreductase; dimerisation domain (lnOR = 1.06; Adj. p = 0.00077) | Alcohol dehydrogenase GroES-like domain (lnOR = 0.87; Adj. p = 0.0017) | ROK family (lnOR = 1.61; Adj. p = 0.0027) | Pyridine nucleotide-disulphide oxidoreductase (lnOR = 0.77; Adj. p = 0.012) | FAD binding domain (lnOR = 1.24; Adj. p = 0.012) |
| H-rich LCDs | Domain of unknown function (DUF3869) (lnOR = 6.02; Adj. p = 0.0054) | Anaphase-promoting complex subunit 4 WD40 domain (lnOR = 2.71; Adj. p = 0.029) | | | | | | | | |
| I-rich LCDs | Clostridium neurotoxin; Translocation domain (lnOR = 3.28; Adj. p = 3.92e-07) | SAC3/GANP family (lnOR = 4.12; Adj. p = 0.0025) | Clostridial neurotoxin zinc protease (lnOR = 2.53; Adj. p = 0.0027) | | | | | | | |
| K-rich LCDs | Fes/CIP4; and EFC/F-BAR homology domain (lnOR = 2.23; Adj. p = 0.0014) | | | | | | | | | |
| L-rich LCDs | Leucine rich repeat N-terminal domain (lnOR = 1.13; Adj. p = 4.59e-06) | Leucine rich repeat N-terminal domain (lnOR = 1.52; Adj. p = 0.0059) | Leucine rich repeat C-terminal domain (lnOR = 1.29; Adj. p = 0.016) | | | | | | | |
| M-rich LCDs | Signal peptide binding domain (lnOR = 4.64; Adj. p = 0.0004) | Domain of unknown function (DUF305) (lnOR = 5.4; Adj. p = 0.0042) | Septin (lnOR = 4.71; Adj. p = 0.014) | NOPS (NUC059) domain (lnOR = 4.6; Adj. p = 0.017) | Multicopper oxidase (lnOR = 3.17; Adj. p = 0.025) | Multicopper oxidase (lnOR = 2.95; Adj. p = 0.044) | | | | |
| N-rich LCDs | Bacterial adhesion/invasion protein N terminal (lnOR = 3.25; Adj. p = 2.92e-08) | Clostridium neurotoxin; Translocation domain (lnOR = 3.3; Adj. p = 1.73e-05) | Clostridial neurotoxin zinc protease (lnOR = 2.93; Adj. p = 2.2e-05) | Duffy binding domain (lnOR = 3.04; Adj. p = 0.00065) | Pectate lyase superfamily protein (lnOR = 2.69; Adj. p = 0.00068) | Clostridium neurotoxin; N-terminal receptor binding (lnOR = 2.49; Adj. p = 0.011) | Alpha-2-macroglobulin MG1 domain (lnOR = 4.33; Adj. p = 0.018) | Nontoxic nonhaemagglutinin C-terminal (lnOR = 3.92; Adj. p = 0.041) | | |

*(Continued)*

**Table 1.** (Continued)

| LCD Type | Annotation 1 | Annotation 2 | Annotation 3 | Annotation 4 | Annotation 5 | Annotation 6 | Annotation 7 | Annotation 8 | Annotation 9 | Annotation 10 |
|---|---|---|---|---|---|---|---|---|---|---|
| P-rich LCDs | Collagen triple helix repeat (20 copies) (lnOR = 1.88; Adj. p = 0.0011) | | | | | | | | | |
| Q-rich LCDs | Retroviral envelope protein (lnOR = 2.18; Adj. p = 4.87e-07) | Cupin (lnOR = 2.02; Adj. p = 3.87e-05) | Fes/CIP4; and EFC/F-BAR homology domain (lnOR = 3.0; Adj. p = 0.00082) | STAT protein; all-alpha domain (lnOR = 4.05; Adj. p = 0.0017) | XPC-binding domain (lnOR = 3.7; Adj. p = 0.005) | Protease inhibitor/seed storage/LTP family (lnOR = 2.58; Adj. p = 0.037) | | | | |
| R-rich LCDs | Helicase conserved C-terminal domain (lnOR = 1.47; Adj. p = 1.62e-08) | Hepatitis C virus NS3 protease (lnOR = 2.42; Adj. p = 0.0025) | CXXC zinc finger domain (lnOR = 2.36; Adj. p = 0.015) | Snurportin1 (lnOR = 3.52; Adj. p = 0.036) | | | | | | |
| S-rich LCDs | Immunoglobulin C1-set domain (lnOR = 1.13; Adj. p = 0.0) | Immunoglobulin V-set domain (lnOR = 0.94; Adj. p = 0.0) | | | | | | | | |
| T-rich LCDs | Immunoglobulin C1-set domain (lnOR = 0.55; Adj. p = 1.52e-07) | B domain (lnOR = 2.69; Adj. p = 5.25e-07) | Prion/Doppel alpha-helical domain (lnOR = 2.9; Adj. p = 2.86e-06) | Immunoglobulin V-set domain (lnOR = 0.4; Adj. p = 0.00061) | Gram-positive pilin backbone subunit 2; Cna-B-like domain (lnOR = 2.59; Adj. p = 0.00074) | Gram-positive pilin subunit D1; N-terminal (lnOR = 2.73; Adj. p = 0.0017) | Urease alpha-subunit; N-terminal domain (lnOR = 1.87; Adj. p = 0.026) | Mur ligase family; catalytic domain (lnOR = 2.02; Adj. p = 0.027) | Mur ligase family; glutamate ligase domain (lnOR = 2.02; Adj. p = 0.027) | Glycosyl hydrolase family 7 (lnOR = 2.21; Adj. p = 0.03) |
| V-rich LCDs | Alcohol dehydrogenase GroES-like domain (lnOR = 1.41; Adj. p = 5.44e-08) | Zinc-binding dehydrogenase (lnOR = 1.31; Adj. p = 0.00016) | Subtilase family (lnOR = 1.44; Adj. p = 0.0034) | Zinc-binding dehydrogenase (lnOR = 1.52; Adj. p = 0.011) | | | | | | |
| W-rich LCDs | Cellulase (glycosyl hydrolase family 5) (lnOR = 4.6; Adj. p = 3.38e-06) | Reverse transcriptase connection domain (lnOR = 6.66; Adj. p = 7.48e-05) | Reverse transcriptase thumb domain (lnOR = 6.48; Adj. p = 9.57e-05) | RNase H (lnOR = 5.6; Adj. p = 0.00045) | Reverse transcriptase (RNA-dependent DNA polymerase) (lnOR = 5.6; Adj. p = 0.00045) | Domain of unknown function (DUF4136) (lnOR = 7.17; Adj. p = 0.011) | Domain of unknown function (DUF1957) (lnOR = 6.66; Adj. p = 0.015) | Glycosyl hydrolase family 57 (lnOR = 5.87; Adj. p = 0.026) | Sortilin; neurotensin receptor 3; (lnOR = 5.71; Adj. p = 0.027) | BNR repeat-like domain (lnOR = 5.71; Adj. p = 0.027) |
| Y-rich LCDs | Scavenger mRNA decapping enzyme C-term binding (lnOR = 4.23; Adj. p = 0.0039) | Scavenger mRNA decapping enzyme (DcpS) N-terminal (lnOR = 4.23; Adj. p = 0.0039) | Peptidase family M3 (lnOR = 4.0; Adj. p = 0.0068) | Phospholipase A2 (lnOR = 2.46; Adj. p = 0.017) | Staphylococcus aureus coagulase (lnOR = 4.92; Adj. p = 0.034) | WxcM-like; C-terminal (lnOR = 3.34; Adj. p = 0.039) | | | | |

these results suggest that each type of LCD may fulfill a specialized structural role in particular tertiary structure families.

## Discussion

LCDs have gained recent attention for their importance in a number of molecular processes. While LCDs in protein sequences have been studied for nearly two decades [22,25,44], a complete understanding of the forms and functions of LCDs is still lacking. It is often asserted that LCDs tend to be intrinsically disordered. On a statistical level this is true, since the majority of LCDs are enriched in polar and charged amino acids (see Fig 2 and [14,15,18]) which, at high local density, tend to promote disordered conformations [24]. However, this is an oversimplification on two levels: 1) LCDs are, in reality, a highly heterogeneous class of domains with distinct physicochemical properties based on LCD sequence and amino acid composition, which results in diverging structural tendencies, and 2) traditional LCD methods do not distinguish between LCDs with only mild compositional enrichment (~25–35% of a single type of amino acid) and those with extreme compositional enrichment (e.g. true homopolymers). Therefore,

both the nature and magnitude of amino acid enrichment within LCDs are effectively lost when LCDs are treated as a single category.

In the present work, we advance the view that LCDs ought to be parsed into sub-types based on the predominant amino acid enriched and its degree of enrichment within each LCD. Analyses by Saqi [45] and Kumari *et al*. [3] indicate that LCDs, when treated as a single class, typically adopt α-helical or coiled conformations in structured proteins. Furthermore, both studies showed that the amino acid compositions of these LCDs differ from the compositions of LCDs in naturally occurring proteins in general protein sequence databases. These observations provided important broad statistics with respect to LCDs as a single class and highlighted the fact that LCDs are not universally disordered. However, they did not specifically dissect the relationships between the compositions of LCDs and their corresponding structural tendencies. Here, we observe that although many types of LCDs are indeed predominantly disordered, a number of LCD sub-classes exhibit substantial tendencies to form structured conformations that are strongly dependent upon the predominant amino acid comprising the LCD. Furthermore, at milder levels of enrichment, even LCD classes typically associated with disorder may retain at least a minor tendency to form ordered conformations (e.g. Q-rich, T-rich, E-rich, R-rich, and K-rich LCDs). For all LCD types, structural tendencies vary as a function of amino acid composition and often change in a dose-dependent manner upon progressive enrichment.

With respect to tertiary structure, certain types of LCDs were preferentially associated with particular protein fold families, suggesting that LCDs may fulfill specific structural roles in some protein folds. Although we were able to map Pfam annotations to each PDB chain, Pfam annotations could not be mapped to precise locations within PDB chain sequences. Therefore, Pfam associations with specific types of LCDs can be interpreted in multiple ways. For example, it is possible that LCDs tend to lie directly within associated Pfam-annotated regions, or that LCDs lying outside of Pfam-annotated regions interact with or dock on the domain.

When compared with traditional secondary structure propensity scales, the average secondary structure content among LCD subtypes exhibited only a moderate or weak correlation. Most secondary structure scales are based predominantly on experiments or statistics that are heavily or exclusively reliant upon observations among high-complexity sequences. Our observations suggest that traditional scales may only weakly capture structural tendencies at the more "unusual" compositional extremes that define LCDs. Perhaps the broader principle (which we are certainly not the first to appreciate) is that secondary structure scales are not static: secondary structure propensities for each of the amino acids, though often represented by a single value, likely change based on local amino acid sequence and composition. By definition, LCDs provide an unusual local environment and are, therefore, no exception to this principle.

Our study also provides ample opportunity for follow-up analyses examining specific aspects of the data in greater detail. While parsing LCDs by amino acid composition enhances the ability to resolve differences between these classes, there may yet exist sub-classes of LCDs within each category which exhibit unique biophysical behavior. For example, while highly charged domains tend to be intrinsically disordered [24], specific patterning of oppositely charged residues can, in some cases, favor alpha helical conformations [46]. Indeed, when we re-analyzed the 90% non-redundant PDB while precluding all windows in which oppositely charged residues co-occur, we observe a slight downward shift in the alpha helix curves for all of the main charged residues (S12 Fig; see Methods for details), suggesting that charged single alpha helices may contribute (albeit only slightly) to the structural tendencies of these classes of LCDs. Therefore, while the analyses presented in this work lay a necessary foundation for

studying the general structural behavior of LCDs, additional layers of detail may aid in the characterization of nuanced sub-classes of LCDs.

It is important to note that, although LCDs are abundant within structured proteins, these LCDs likely constitute only a small fraction of all LCDs found in native proteins (refer back to Figs 1 and 2). The secondary structure content of LCDs observed in structured proteins might necessarily be skewed toward structured conformations and, therefore, may not reflect the tendencies of each type of LCD globally. Furthermore, it is conceivable that structural studies are biased at the experimental level: 1) LCDs might in some cases be eliminated *a priori* from protein sequences due to the pervasive perception that they tend to be disordered, 2) solution conditions used in structural studies might favor or disfavor ordered conformations for certain types of LCDs, and 3) many types of LCDs may adopt structures contingent upon interaction with specific binding partners. Therefore, at present, it is difficult to evaluate whether incorporating composition-dependence into secondary structure propensity scales would improve *de novo* predictions of secondary structure based on sequence alone.

In conclusion, our results highlight the abundance of LCDs within structured proteins, the unique structural tendencies of LCDs, and the differences in structural tendencies across LCDs of distinct types within the PDB proteome.

## Methods

### Data acquisition and processing

Protein sequences as well as corresponding secondary structure and intrinsic disorder annotations for all protein chains found in the Protein Data Bank (PDB) were downloaded from https://www.rcsb.org/pdb/static.do?p=download/http/index.html on 6/19/2019. To reduce bias due to redundancy in the PDB, a set of PDB IDs clustered based on a 90% homology threshold were obtained from http://www.rcsb.org/pages/download/ftp on 6/19/2019 (additional follow-up analyses were performed on a set of PDB sequences with <40% homology, downloaded on 11/8/19). For each cluster, the first PDB ID (which is pre-sorted based on the quality of the structure deposited in the PDB) was retained in the "non-redundant" set. To further eliminate LCD biases due to N- or C-terminal tags (e.g. HA-tags, polyHis tags, etc.), tags were trimmed from PDB chain sequences if the tag sequence overlapped with the first 20 or final 20 residues within the sequence (see S5 File for full list of tags and corresponding tag sequences included in search criteria). The human and yeast reference proteomes (Uniprot IDs UP000005640 and UP000002311, respectively) were downloaded from https://www.uniprot.org/ on 10/5/2019.

### Calculation of Shannon entropy

Shannon entropy, which is implemented in the SEG algorithm to identify LCDs [25], was calculated as indicated in [18]. Briefly, the Shannon entropy corresponding to each 12aa subsequence was calculated as:

$$SE = -\sum_{i=1}^{N=20} \frac{n_i}{L} \left( log_2 \frac{n_i}{L} \right) \tag{1}$$

where $N$ represents the size of the residue alphabet ($N = 20$, for the canonical amino acids), $n_i$ is the number of occurrences of the $i$th residue within the given subsequence of length $L = 12$. LCDs (defined as having SE $\leq$ 2.2 bits) were then assigned to a LCD category based on the most frequent amino acid within the subsequence (i.e. A-rich LCDs, C-rich LCDs, etc.).

## Local amino acid composition algorithm

To examine the distribution of protein segments as a function of composition and window size for the entire non-redundant PDB, all PDB chain protein sequences were exhaustively scanned using window sizes ranging from 10 amino acids to 100 amino acids. For each amino acid, the percent composition for each subsequence was calculated and binned based on the composition for the amino acid of interest. All subsequences containing a non-canonical amino acid were excluded from analyses.

To examine relationships between amino acid composition and secondary structure tendencies, a 12aa scanning window (which corresponds to the default window size implemented in the SEG algorithm [25,44]) was used to exhaustively scan each PDB chain sequence. The mean fraction of each secondary structure type was calculated independently for each combination of amino acid and residue count (i.e. composition). Within each dataset corresponding to one amino acid, these mean fraction values for each secondary structure type were linearly interpolated and plotted as a dynamic bar chart to generate S1 Movie. Additionally, to compare with LCDs defined using traditional methods, the mean fraction of each secondary structure type was calculated for each class of LCDs defined by Shannon entropy, as indicated above.

## Regression analyses comparing observed secondary structure content to established secondary structure propensity scales

For each set of highly-enriched LCDs analyzed, the fraction of the defining amino acid (i.e. the amino acid constituting ≥50% of each LCD) annotated as α-helix or β-sheet was calculated, and those values were plotted in a pairwise fashion against each of the α-helix or β-sheet propensity scales, respectively. The same procedure was performed for non-LCD regions as well: specifically, for each amino acid, the fraction of residues in α-helices or β-sheets (and not within a highly-enriched LCD of that amino acid type) was calculated and similarly plotted against the α-helix or β-sheet propensity scales, respectively. Ordinary least squares regressions were performed using the Python statsmodels package. Residual values were calculated from the resulting regression lines and the associated fraction of residues in α-helices or β-sheets for each amino acid among LCDs and among non-LCD regions. The absolute value of the correlation coefficients for all pairwise comparisons between secondary structure propensity scales, LCD regions, and non-LCD regions (S6 Fig) were calculated using the SciPy (Python package) pearsonr correlation function. Correlation matrices in S6 Fig were sorted based on average correlation coefficient. Since proline is often an outlier or absent altogether among established secondary structure propensity scales, it was excluded from all regression analyses (as was glycine specifically from the Kim & Berg [42] β-sheet propensity scale, as determined by examination of initial regression plots of the non-LCD-derived statistics). Additionally, LCD classes with fewer than 50 total counts of the defining amino acid were excluded from analyses due to small sample sizes.

## Statistical analysis of Pfam annotations associated with LCDs

Pfam protein fold annotations were obtained from the PDBfam database ([43]; http://dunbrack2.fccc.edu/protcid/pdbfam/Download.aspx) on 6/25/2019. For each LCD type, Pfam annotations were mapped to each PDB chain in the LCD category. The frequency of each Pfam annotation found in each LCD category was then compared to the overall frequency of the Pfam annotation within the non-redundant PDB proteome using a two-sided Fisher's exact test. Adjusted *p*-values were calculated within each LCD category using the Holm-Šidák

method (implemented in the statsmodels package available in Python) to account for multiple hypothesis testing. Additionally, to evaluate the degree of enrichment or depletion of each Pfam annotation associated with LCDs, the odds ratio was calculated as:

$$OR_A = \left[\frac{f_{LCD}}{1 - f_{LCD}}\right] \Big/ \left[\frac{f_{PDB}}{1 - f_{PDB}}\right] \tag{2}$$

where $f_{LCD}$ represents the frequency of the Pfam annotation among a given class of LCDs, and $f_{PDB}$ represents the frequency of the Pfam annotation among all chains in the non-redundant PDB proteome.

## Supporting information

**S1 Movie. Progressive development of secondary structure tendencies as a function of compositional enrichment.** For each amino acid, the mean fraction values corresponding to each type of secondary structure were linearly interpolated across all residue count bins. Each stacked bar represents the mean secondary structure content for each secondary structure type corresponding to all peptides with the indicated amino acid composition. For example, in the first frame (titled "0 Residues of Interest"), the "A" stacked bar represents the mean secondary structure content corresponding to all peptides with 0 alanine residues (note that at this stage, all stacked bars are quite similar and effectively represent rough approximations of the mean secondary structure profile for the entire PDB proteome). As the movie progresses, the stacked bars change to reflect the secondary structure content as the composition increases for each of the amino acids on the x-axis. To continue the example, "1 Residue of Interest" indicates the secondary structure profile corresponding to all peptides with 1 alanine for the "A" stacked bar, "2 Residues of Interest" represents the secondary structure profile corresponding to all peptides with 2 alanines, etc. Short interframe intervals are indicative of interpolated values, whereas longer pauses occur as each integer value for "X Residues of Interest" is reached.
(MP4)

**S1 Fig. Sample sizes vary as a function of amino acid composition.** For each amino acid, the bar plot indicates the sample sizes in each residue count bin corresponding to Fig 4.
(TIF)

**S2 Fig. Secondary structure tendencies as a function of composition among a non-redundant PDB dataset with <40% homology are nearly identical to those observed with a 90% homology threshold.** A set of PDB sequences with <40% sequence homology was analyzed with the same computational procedure (Fig 3) applied to the <90% homology PDB dataset. Compare with Fig 4 in the main text. It is worth noting that the PDB is updated regularly, and archived versions of the 40% non-redundant PDB are not available: therefore, the 40% non-redundant PDB proteome contains the structures (and sequences) of new proteins solved since our initial download of the 90% non-redundant PDB proteome. The full list of PDB chain IDs included in the 40% non-redundant PDB proteome is contained in S2 File.
(TIF)

**S3 Fig. Structural tendency profiles observed for classically-defined LCDs are not substantially altered when using a lower sequence homology threshold (40%).** Classically-defined LCDs were identified among the 40% non-redundant PDB dataset in the same manner as described for Fig 5.
(TIF)

**S4 Fig. Observed fraction of the LCD-defining residue in α-helices among highly-enriched LCDs exhibits a moderate correlation with α-helix propensity scales.** Scatter plots indicate all pairwise comparisons between the fraction of the LCD-defining residue in α-helices among each type of highly-enriched LCD and values from established α-helix propensity scales. Each shaded band indicates the 95% confidence interval around the regression line.
(TIF)

**S5 Fig. Observed fraction of the LCD-defining residue in β-sheets among highly-enriched LCDs exhibits a moderate-to-weak correlation with β-sheet propensity scales.** Scatter plots indicate all pairwise comparisons between the fraction of the LCD-defining residue in β-sheets among each type of highly-enriched LCD and values from established β-sheet propensity scales. Each shaded band indicates the 95% confidence interval around the regression line.
(TIF)

**S6 Fig. α-helix and β-sheet content among highly-enriched LCDs diverges from secondary structure propensity scales.** All pairwise comparisons between secondary structure propensity scales, as well as the fraction of each amino acid corresponding to each secondary structure type among highly-enriched LCDs and non-LCD regions, were performed for α-helix (A) and β-sheet (B) propensity scales (see also S4 and S5 Figs for each pairwise correlation plot). Heatmap intensities and corresponding values indicate the absolute value of the Pearson correlation coefficient for each comparison. To indicate general ranking with respect to the overall degree of correlation with all other α-helix or β-sheet propensity scales, the heatmaps were pre-sorted based on the average absolute correlation coefficient in descending order (top-to-bottom, and left-to-right).
(TIF)

**S7 Fig. Observed fraction of residues among non-LCDs in α-helices exhibits a strong correlation with α-helix propensity scales.** Scatter plots indicate all pairwise comparisons between the fraction of each residue in α-helices among the complementary set of non-LCD regions for each class of LCD and values from established α-helix propensity scales. Each shaded band indicates the 95% confidence interval around the regression line.
(TIF)

**S8 Fig. Observed fraction of residues among non-LCDs in β-sheets exhibits a strong correlation with β-sheet propensity scales.** Scatter plots indicate all pairwise comparisons between the fraction of each residue in β-sheets among the complementary set of non-LCD regions for each class of LCD and values from established β-sheet propensity scales. Each shaded band indicates the 95% confidence interval around the regression line.
(TIF)

**S9 Fig. Dose-dependent relationships between amino acid composition and divergence from α-helix propensity scales.** For each residue count bin, the fraction of the LCD-defining amino acid in α-helices was calculated separately for each LCD class. Pairwise regression analyses were performed between each α-helix propensity scale and the fraction of α-helical residues among highly-enriched LCDs. From the regression analyses, the residual values for each LCD class (i.e. each amino acid) were averaged across all α-helix propensity scales. This process was repeated independently for each residue count bin, and the resulting mean residual values are indicated in the figure above. Additionally, the size of each point (as well as the opacity of each point and the opacity of the preceding line segment) reflects the sample size for each residue count bin for each LCD class (i.e. the number of protein regions that were parsed

into each bin based on the composition of the amino acid indicated in each subplot title).
(TIF)

**S10 Fig. Dose-dependent relationships between amino acid composition and divergence from β-sheet propensity scales.** Mean residual values were calculated from pairwise regression analyses between observed fraction of LCD-defining residues in β-sheets among highly-enriched LCDs and each of the β-sheet propensity scales. Regression analyses, calculation of the residuals, and plotting were performed as indicated in the S9 Fig legend.
(TIF)

**S11 Fig. Highly-enriched LCDs do not exhibit strong subsidiary preferences for other amino acids.** Heatmap indicating the average frequency of all residues calculated from the highly-enriched LCD sequences within each LCD category.
(TIF)

**S12 Fig. α-helix peaks are only mildly affected when precluding the co-occurrence of oppositely charged residues.** Secondary structure proportions were re-calculated for each of the main charged residues while eliminating all windows that contain an oppositely charged residue (see Methods). In all cases, the α-helix peaks exhibit a slight downward shift (compare with Fig 4), suggesting that charged single α-helices constitute only a minor contribution to the overall shape and magnitude of the α-helix curves for charged residues.
(TIF)

**S1 File. PDB chain IDs included in the 90% non-redundant PDB dataset.** The original set of non-redundant PDB sequences (at the 90% sequence identity level) was further filtered to exclude all PDB chains associated with membrane proteins (as indicated in the Methods section). S1 File contains the final set of PDB chain IDs used for all analyses pertaining to the 90% non-redundant dataset.
(TXT)

**S2 File. PDB chain IDs included in the 40% non-redundant PDB dataset.** Membrane proteins were filtered from the downloaded set of non-redundant PDB sequences (at the 40% sequence identity level). The final set of PDB chain IDs used for all analyses pertaining to the 40% non-redundant dataset are contained in S2 File.
(TXT)

**S3 File. Supplemental discussion for S9 and S10 Figs.**
(DOCX)

**S4 File. Statistical analyses of Pfam annotations associated with each type of LCD.** All Pfam annotations associated with each LCD category (separate tabs) are indicated, along with their frequencies among the LCD category and PDB proteome, their degrees of enrichment/depletion (lnOR), and their associated probability statistics (*p*-value and Holm-Šidák corrected *p*-value).
(XLSX)

**S5 File. Common protein expression/purification tags trimmed from protein termini prior to analyses.**
(XLSX)

## Acknowledgments

We would like to thank Dr. Olve Peersen for critical suggestions regarding analytical approach.

## Author Contributions

**Conceptualization:** Sean M. Cascarina, Eric D. Ross.

**Data curation:** Sean M. Cascarina.

**Formal analysis:** Sean M. Cascarina, Mikaela R. Elder.

**Funding acquisition:** Eric D. Ross.

**Investigation:** Sean M. Cascarina.

**Methodology:** Sean M. Cascarina, Mikaela R. Elder, Eric D. Ross.

**Project administration:** Sean M. Cascarina, Eric D. Ross.

**Resources:** Eric D. Ross.

**Software:** Sean M. Cascarina.

**Supervision:** Sean M. Cascarina, Eric D. Ross.

**Validation:** Sean M. Cascarina, Mikaela R. Elder.

**Visualization:** Sean M. Cascarina.

**Writing – original draft:** Sean M. Cascarina.

**Writing – review & editing:** Sean M. Cascarina, Mikaela R. Elder, Eric D. Ross.

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
