## [Decision Letter · Decision Letter 0]

7 Nov 2019

Dear Dr Cascarina,

Thank you very much for submitting your manuscript 'Atypical Structural Tendencies Among Low-Complexity Domains in the Protein Data Bank Proteome' for review by PLOS Computational Biology. Your manuscript has been fully evaluated by the PLOS Computational Biology editorial team and in this case also by independent peer reviewers. The reviewers appreciated the attention to an important problem, but raised some substantial concerns about the manuscript as it currently stands. While your manuscript cannot be accepted in its present form, we are willing to consider a revised version in which the issues raised by the reviewers have been adequately addressed. We cannot, of course, promise publication at that time.

Sincerely,

Björn Wallner

Associate Editor

PLOS Computational Biology

Arne Elofsson

Deputy Editor

PLOS Computational Biology

[LINK]

Reviewer's Responses to Questions

**Comments to the Authors:**

Reviewer #1: This article presents some analysis of the relationship between 'low complexity' and Protein Data

Bank structural preferences. This is an interesting work, although I have some points for revision.

* The data set of protein chains is constructed using a sequence identity threshold of 90% to remove redundancy. This is high. Usually, people would use

something lower like 40% or 30%. Can the authors

justify such a high threshold? They should show that the at least some of the results are not affected

by such a high threshold.

* There is some discussion about the performance of secondary structure propensity scales for low complexity sequences (centred around Figs S1, S2, etc.). There does not seem to be a comparative discussion of the performance of propensity scales for non-LCD segments of protein structures as well. I think

the authors should add this comparison and discuss the

differences.

Reviewer #2: Please see attached comments.

**Have all data underlying the figures and results presented in the manuscript been provided?**

Reviewer #1: Yes

Reviewer #2: Yes

PLOS authors have the option to publish the peer review history of their article (what does this mean?). If published, this will include your full peer review and any attached files.

Reviewer #1: No

Reviewer #2: No

---

## [Decision Letter · Decision Letter 1]

23 Dec 2019

Dear Dr Cascarina,

We are pleased to inform you that your manuscript 'Atypical Structural Tendencies Among Low-Complexity Domains in the Protein Data Bank Proteome' has been provisionally accepted for publication in PLOS Computational Biology.

In the meantime, please log into Editorial Manager at https://www.editorialmanager.com/pcompbiol/, click the "Update My Information" link at the top of the page, and update your user information to ensure an efficient production and billing process.

One of the goals of PLOS is to make science accessible to educators and the public. PLOS staff issue occasional press releases and make early versions of PLOS Computational Biology articles available to science writers and journalists. PLOS staff also collaborate with Communication and Public Information Offices and would be happy to work with the relevant people at your institution or funding agency. If your institution or funding agency is interested in promoting your findings, please ask them to coordinate their releases with PLOS (contact ploscompbiol@plos.org).

Thank you again for supporting Open Access publishing. We look forward to publishing your paper in PLOS Computational Biology.

Sincerely,

Björn Wallner

Associate Editor

PLOS Computational Biology

Arne Elofsson

Deputy Editor

PLOS Computational Biology

Reviewer's Responses to Questions

**Comments to the Authors:**

Reviewer #1: My previous comments have been addressed.

Reviewer #2: I thank the authors for their detailed response to the reviewer comments, and am fully satisfied by the changes made.

**Have all data underlying the figures and results presented in the manuscript been provided?**

Reviewer #1: Yes

Reviewer #2: Yes

PLOS authors have the option to publish the peer review history of their article (what does this mean?). If published, this will include your full peer review and any attached files.

Reviewer #1: No

Reviewer #2: No

---

## [Editor Report · Acceptance letter]

21 Jan 2020

PCOMPBIOL-D-19-01766R1 

Atypical Structural Tendencies Among Low-Complexity Domains in the Protein Data Bank Proteome

Dear Dr Cascarina,

I am pleased to inform you that your manuscript has been formally accepted for publication in PLOS Computational Biology. Your manuscript is now with our production department and you will be notified of the publication date in due course.

With kind regards,

Matt Lyles
